# Circulating microRNAs for Early Diagnosis of Ovarian Cancer: A Systematic Review and Meta-Analysis

**DOI:** 10.3390/biom13050871

**Published:** 2023-05-22

**Authors:** Nanna Lond Skov Frisk, Anja Elaine Sørensen, Ole Birger Vesterager Pedersen, Louise Torp Dalgaard

**Affiliations:** 1Department of Science and Environment, Roskilde University, Universitetsvej 1, 4000 Roskilde, Denmark; 2Department of Clinical Immunology, Zealand University Hospital, Køge, Ringstedgade 77B, 4700 Næstved, Denmark; 3Department of Clinical Medicine, Faculty of Health and Medical Science, University of Copenhagen, Blegdamsvej 3, 2200 Copenhagen N, Denmark

**Keywords:** microRNA, ovarian cancer, biomarker, diagnostics, meta-analysis, systematic review, miR-21, miR-205, miR-106, miR-328, miR-26, miR-141, miR-200c, miR-200b, miR-429

## Abstract

In this study, we conducted a systematic review and meta-analysis to summarize and evaluate the global research potential of different circulating miRNAs as an early diagnostic biomarker for OC. A systematic literature search for relevant studies was conducted in June 2020 and followed up in November 2021. The search was conducted in English databases (PubMed, ScienceDirect). The primary search resulted in a total of 1887 articles, which were screened according to the prior established inclusion and exclusion criteria. We identified 44 relevant studies, of which 22 were eligible for the quantitative meta-analysis. Statistical analysis was performed using the Meta-package in Rstudio. Standardized mean differences (SMD) of relative levels between control subjects and OC patients were used to evaluate the differential expression. All studies were quality evaluated using a Newcastle–Ottawa Scale. Based on the meta-analysis, nine miRNAs were identified as dysregulated in OC patients compared to controls. Nine were upregulated in OC patients compared to controls (miR-21, -125, -141, -145, -205, -328, -200a, -200b, -200c). Furthermore, miR-26, -93, -106 and -200a were analyzed, but did not present an overall significant difference between OC patients and controls. These observations should be considered when performing future studies of circulating miRNAs in relation to OC: sufficient size of clinical cohorts, development of consensus guidelines for circulating miRNA measurements, and coverage of previously reported miRNAs.

## 1. Introduction

Ovarian cancer (OC) is one of the foremost causes of gynecological cancer deaths [1]. Symptoms are often non-specific, thereby obstructing early diagnosis, which means that most women present with advanced-stage disease [2]. OC patients have a median age of 63 years when diagnosed; however, women with high-risk factors may be diagnosed earlier. Family history is a substantial risk factor; a woman with a first-degree relative with a history of the disease has a fourfold increased risk, whereas a second-degree relative having OC confers a twofold increased risk [3,4]. Only 42% of affected women live past five years after diagnosis [1].

Screening is applying one or more tests to an asymptomatic at-risk population to detect a specific disease at an earlier and more curable stage. Different OC screening approaches were applied in the UK and USA. In the USA, a screening program for prostate, lung, colorectal, and OC was tested. The results of the screening tests were discouraging because performing an annual screening using cancer antigen 125 (CA125) and transvaginal sonography did not markedly reduce the mortality rate of OC [5]. Similar findings were observed in the UK Collaborative Trial of Ovarian Cancer (UKCTOCS) [6]. A biomarker is a naturally occurring molecule that can identify a pathological or physiological process and disease [7]. In OC, circulating CA125 protein and human epididymis protein 4 (HE4) are currently applied as biomarkers [8]; however, they lack sensitivity and are only detected at high levels in advanced stages in patients with OC. Accordingly, CA125 is only detected in 50% of patients with stage I OC [9,10]. Therefore, new biomarkers with higher sensitivity and specificity for OC would be highly advantageous because they would enable earlier diagnosis and more timely treatment of OC with the possibility of improved treatment results.

Circulating microRNAs (miRNAs) may constitute novel biomarkers for OC. MiRNAs are a class of small non-coding (nc)RNAs having an average of 22 nucleotides in length and were discovered by Lee, Feinbaum, and Ambros in 1993 [11]. Intracellularly, their primary role is the downregulation of the expression of their target genes via interaction with the 3′ untranslated region (3’ UTR) of target mRNAs [12,13,14]. MiRNAs were detected in almost all body fluids [12,13]. Furthermore, they are secreted from cells into the bloodstream in a stable and reproducible form in serum and plasma because they are protected from degradation by either being bound to specific RNA-binding proteins or encapsulated in exosomes [15,16]. More than 2500 miRNAs were identified in the human genome, and miRNAs regulate at least 30% of protein-coding genes [13,17,18]. MiRNAs were quantified in multiple studies of OC patients to identify a novel diagnostic or prognostic biomarker for OC. The need for an early novel diagnostic biomarker is substantial, and the literature is not consistent regarding the miRNAs reported associated with OC. 

A major obstacle to using qPCR-based biomarkers in clinical settings is the lack of technical consistency. Limitations are also related to the lack of agreed-upon reference values, the poor harmonization of the study populations, small individual study sizes and challenges in commercial, academic, and medical collaboration. For instance, few possible indications were effectively converted into clinical practice despite the thousands of noncoding RNA (ncRNA)-based biomarker studies that were published so far, primarily because the results of these studies are not reproducible [19].

This systematic review summarizes and integrates previous findings to provide an overview of all available literature on circulating miRNAs in OC. Furthermore, the accompanying meta-analysis is the first to provide a quantitative and qualitative assessment of miRNAs related to the diagnostic properties of OC.

## 2. Materials and Methods

### 2.1. Electronic Search and Study Selection

This analysis was conducted according to the PRISMA guidelines [20]. The review was conducted following the published Prospero protocol: CRD42022237812 [21]. Independent searches of PubMed and ScienceDirect databases were performed. The last search was completed in February 2023. Reviews, case reports, conference abstracts, and non-published data were excluded. The following search string was used: (((Ovarian cancer) AND (biomarker)) AND (microRNA OR miRNA)) AND (plasma OR serum). Journal articles or reviews describing miRNAs and OC were further examined manually to include additional studies. Only original studies published in English were included. Covidence.org was used as a reviewing tool [22]. Studies were included if they: (1) investigated miRNA as a diagnostic biomarker for OC and OC subtypes, (2) the clinical samples were derived from the patient’s plasma or serum, and (3) there was an analysis of the association between miRNA levels and OC. Studies on other cancer types, ovarian tissue, drug resistance, or chemotherapy were excluded. Two researchers (NLSF and AES) independently searched the databases, reviewed abstracts, and were blinded to each other’s results. Agreement upon selected abstracts was reached afterward, and studies were then screened full text for inclusion. All databases were screened from inception up to February 2023.

### 2.2. Data Extraction

In a pre-designed table, the following data were extracted: first author, publication year, characteristics of the recruited OC patients and controls, the miRNA expression profile platform, number of participants, bio-fluid type, and quantitative data such as *p*-values (if available), and the degree of miRNA up-or down-regulation and fold changes (if available). If not given in the original manuscript, quantitative measures of miRNA expression were extracted from plots using the tool Web Plot Digitizer [23]. All data were recalculated as relative fold-change between control subjects and OC patients, with control subjects being set to 1 (100%) (S1). Information about miRNA isoform -3p and -5p arms was extracted and can be found in the Appendix A. For some of the articles, the information of 3p and 5p arms of the investigated miRNAs was not given; in these cases, the doubts were resolved using the miRNA database miRbase.org [24].

### 2.3. Quality Assessment

The quality of studies was evaluated using a modified Newcastle–Ottawa quality assessment Scale (NOS) (S1) [25]. The NOS is a newer scale for assessing the quality of non-randomized studies in meta-analyses. The advantage of the NOS scale is that it relates to non-randomized studies, which most biomarker studies belong to, whereas another popular appraisal tool, the Cochrane risk of bias tool vs. 2, is more relevant for randomized studies [25]. We evaluated eight properties in total, giving a possible maximal score of eight stars, one for each assessed property. The studies were scored on three criteria: selection, comparability, and method, with sub-questions. Point-scoring studies fulfill the answer marked with an asterisk (*). A sum of the points was calculated and visualized using a heat map indicating the combined quality of the study: Darker blue indicates higher quality and darker red indicates a lower degree of NOS criteria fulfillment (S3). 

### 2.4. Data Handling, Statistical Analysis, and Meta-Analysis Methodology

If SDs were not directly given in included studies, these were calculated from the given SEM values or quartile fractions and the number of subjects. To compare the relative miRNA expression across studies, we calculated the relative miRNA expression values based on mean and SD-values for mean, SD, confidence intervals, and corresponding fold changes between groups. The relative values of mean and SD for both the OC and the control group were used as input for a meta-analysis using a fixed-effect model. The fixed-effect model was chosen over a random-effect model because none of the individual miRNAs studied was investigated in 10 or more studies. The statistical program R and the Meta package were used to run the fixed-effect meta-analysis for the selected miRNAs [26]. Statistical heterogeneity was assessed using the t2 and I2 tests. An I2 ≥ 65% was considered a violation of the homogeneity assumption. For a miRNA to be included in the meta-analysis, the miRNA had to be investigated and data extractable from three or more different studies. Biases were evaluated using Cook’s distance and funnel plots. R-studio and GraphPad PRISM were used to construct figures. *p*-Values < 0.05 were considered significant. The PRISMA checklist and the PRISMA abstract checklist are enclosed as Appendix A.

## 3. Results

### 3.1. Search Results

The search of the two databases yielded 1884 potential studies; three more studies were found by hand search. After removing duplicates, 1866 titles and abstracts were screened by two independent researchers yielding 128 studies to be considered for inclusion. After a full-text review, 48 studies remained for qualitative analysis [2,27,28,29,30,31,32,33,34,35,36,37,38,39,40,41,42,43,44,45,46,47,48,49,50,51,52,53,54,55,56,57,58,59,60,61,62,63,64,65,66,67,68,69,70,71,72,73]. Out of the 48 studies, 23 were suitable for quantitative meta-analysis. Twenty-three studies were not included in the quantitative meta-analysis analysis following the data extraction because the identified miRNAs were not identified in three or more studies, their data were only displayed with AUC values or in ROC curves, or quantifiable data were not obtainable (Figure 1). Study characteristics of the studies included in the meta-analysis are summarized in Table 1, and the complete list of studies can be found in Appendix A.

### 3.2. Study Characteristics and Quality Assessment

Forty-eight articles were included in the review based on the literature screening and quantitative data extraction. Using the NOS score, the studies included in the systematic review were evaluated, two studies scored three stars [30,45], five studies scored four stars, 12 studies scored five stars, 18 studies scored six stars, eight studies scored seven stars, and only four studies scored a total of eight stars (Appendix A). A total of 131 different miRNAs were investigated in the 48 published studies. However, the majority of the investigated miRNAs were identified in only one or two studies (Figure 2). The 48 studies included 3387 OC patients, 3461 healthy women, and 475 women with benign cysts. A general observation in multiple studies was that the selection of controls was not described. If the selection process was described, most of the selected controls were healthy volunteers for a health check-up, or patients with benign cysts included when scheduled for surgery. Multiple studies did not report whether the malignant and control samples were matched for age. Not all studies included in the review had data available for comparison in the meta-analysis, as some only reported fold changes or receiver operator analysis (ROC) AUC curves [28,32,42,44,46,48,50,61,66,67,69,70,71,72]. Additionally, studies were nationally skewed; 58% of the studies were performed in Asia and 27% in Europe, 12% in the USA, and two studies (4%) in Australia, which means that none of the 48 studies were performed in South American or African ethnic populations. Most studies defined the subtype of OC investigated; the most common subtype investigated was epithelial ovarian cancer (EOC), with 40% of studies. Other subtypes investigated were endometriosis-associated ovarian carcinoma (EAOC) (2%), serous ovarian cancer (SOC) (12%), and high-grade serous ovarian cancer (HGSOC) (8%). However, a large number of the studies did not specify the subtype of OC (40%). Twenty-eight out of the 48 studies measured CA125 in the datasets, 21 of the studies used the CA125 data to investigate the miRNAs diagnostic abilities using ROC curves. HE4 was less frequently investigated; six out of the 48 studies investigated HE4. Three studies found CA125 to have a AUC value below the investigated miRNA(s) [30,70,75]. However, all of the studies found that a combination of CA125 and their investigated miRNA(s) had a higher AUC value than CA125 or the miRNA(s) alone [30,33,34,39,42,45,52,65,69,76]. Only two of the studies used HE4 for ROC curves both alone and in combination with the investigated miRNA(s) [52,65].

### 3.3. Evaluation of Pre-Analytical Factors

A majority of studies used serum (69%); the rest used plasma samples. Most studies using plasma failed to report the anticoagulant used; among those who reported, EDTA was the most commonly used anticoagulant (Appendix A). A total of 52% of the studies used a single-step centrifugation process for plasma/serum separation, 19% used two-step centrifugation, and 29% did not provide information about the separation process (Figure 3). For RNA extraction, the studies used different commercially branded reagents such as Trizol (15%) or kits; for example, miRVANA (19%) or miRNeasy (27%). Fifty-four percent used a hypothesis-free method, meaning using qRT-PCR microarrays or next-generation sequencing to identify possibly interesting miRNAs in a screening phase, followed by a specific target qRT-PCR validation phase (Figure 3). Some studies were hypothesis-based (44%) with a priori chosen miRNAs for measurements.

### 3.4. Quantitative Meta-Analysis of Circulating miRNAs in Relation to OC

Based on the data extraction, 26 miRNAs were identified as associated with OC in three or more studies. However, only 13 of the miRNAs had variance estimates given, which enabled inclusion for meta-analyses. The 13 miRNAs were: miR-21-5p, -26-5p, -93-5p, -106b-5p, -125b-5p, -141-5p, -145-5p, -205-5p, -200a-3p, -200b-3p, -200c-3p, -328-5p, and -429-3p. The most frequently investigated miRNA was miR-200c-3p, investigated in 13 original studies. Some of these studies [32,42,50,66,70,72], however, did not report variance estimates and, therefore, not all of the 13 studies were included in the subsequent quantitative meta-analyses.

### 3.5. Meta-Analysis of miRNAs

Thirteen forest plots were generated, of which eight showed a statistically significant association between the miRNA and OC (Figure 4 and Figure 5). The miRNAs that remained statistically different in the meta-analysis were: miR-21-5p, -125-5p, -141-5p, -145-5p, -205-5p, -200b-3p, -200c-3p, and -328-5p. All six were found to be consistently increased in circulation in OC patients (Figure 4 and Figure 5). However, for most meta-analyses, the miRNA prediction interval indicated low predictive power of the miRNA to detect OC due to large variability within and between studies. The miRNA with the highest identified fold change was miR-145-5p, which was determined to have a 2.98-fold difference between OC patients and controls (*p* < 0.01; the standardized mean difference (SMD): 1.98; 95% CI: [1.63; 2.32]) (Figure 4A). MiR-205-5p, miR-21-5p, and miR-328-5p were also strongly associated with OC with a 2.46-fold (MD: 1.46, 95% CI: [1.25; 1.68]) (Figure 4B), a 1.88-fold (SMD: 0.88, 95% CI: [0.69; 1.07]) (Figure 4C), and a 1.56-fold (SMD: 0.56, 95% CI: [0.32; 0.81]) (Figure 4D) higher level in OC patients as compared to controls, respectively. Oliveira et al. 2019 and Resnick et al. 2009 also investigated miR-21-5p, and both found in the meta-analysis that circulating miR-21-5p was upregulated in OC patients compared to controls [42,46] (Appendix A). The association between miR-125-5p and OC was statistically significant in the meta-analysis, with a 1.30-fold increase (SMD: 0.30, 95% CI: [0.01; 1.59]) in OC patients compared to controls (Figure 4E). The association between miR-93-5p and OC was not statistically significant in the meta-analysis, as two studies showed downregulation of miR-93-5p in OC patients. In contrast, two other studies found that miR-93-5p was upregulated in OC patients (Figure 4F) [46,50]. A study by Resnick et al. 2009 identified miR-93-5p to be upregulated in OC patients compared to controls. However, the data were not extractable for the meta-analysis [46]. Circulating miR-26-5p was downregulated in two out of three OC studies. The association between miR-26-5p and OC was that low levels were associated with a diagnosis of OC; however, no overall significant decrease was found in OC patients (SMD: −0.19; 95% CI: [−0.39; 0.01]) (Figure 4G). The study by Penyige et al. 2019 previously reported miR-26-5p to be downregulated in stage IV OC but upregulated in stage I and III OC [44] (Appendix A). Circulating miR-106b-5p was not statistically significantly associated with OC (Figure 4H).

Four out of five miRNAs within the miR-200 family were increased in OC compared to controls. MiR-200c-3p was the miRNA with the highest identified fold-change, which was determined to have a 1.81-fold difference between OC patients and controls (SMD: 0.81; 95% CI: [0.66; 0.97]) (Figure 5A). MiR-200c-3p was closely followed by miR-141-5p with a fold-change of 1.53 (SMD: 0.53; 95% CI: [0.31; 0.75]) (Figure 5B). Lastly, miR-200b-3p and -200a-3p were significantly increased in OC patients compared to controls with 1.20-fold (SMD: 0.20; 95% CI: [0.02; 0.38]) (Figure 5C) and 1.16-fold (SMD: 0.16; 95% CI: [0.01; 0.31]) (Figure 5D). The last miRNA (miR-429-3p) was not identified to be statistically significant expressed in OC patients (SMD: 0.17; 95% CI: [−0.00; 0.35]) (Figure 5D). 

For three different miRNAs, it was possible to perform subgroup analysis regarding sample types: serum and plasma. MiR-200c-3p measured in serum was found to be statistically significantly increased OC patients compared to controls (SMD: 1.29; 95% CI: [1.08; 1.51], *p* < 0.01) (Figure 6A). The subgroup analysis of plasma miR-200c-3p showed a 1.27-fold increase in OC (SMD: 0.27; 95% CI: [0.05; 0.50], *p* < 0.05) (Figure 6C). Thus, miR-200c-3p was overall increased in OC regardless of whether the miRNA was measured in serum or plasma. MiR-205-5p was measured in plasma in three studies and showed overall a 1.78-fold increase in OC patients compared to controls (SMD: 0.78; 95% CI: [0.53; 1.04], *p* < 0.01) (Figure 6B). MiR-200a-3p measured in serum, without the single plasma-based study, was no longer increased in OC (SMD: 0.05; 95% CI: [−0.16; 0.25]) (Figure 6D).

A Cook’s distance analysis did not identify influential outliers. The summary results, including effect size estimates, confidence intervals, and *p*-values, remained consistent when influential studies were excluded from the analysis. The funnel plots were found to be asymmetric (Appendix A), suggesting that publication bias or other sources of heterogeneity may have been present. The plot showed an asymmetric distribution of studies, with few smaller studies and estimations that were less accurate in one area of the plot. This implies that the outcomes of the meta-analysis may have been impacted by publication bias, selective reporting of studies, or other causes. When interpreting the results, caution should be used. The heterogeneity between studies in the meta-analyses was high, with 97% (Figure 4A), 99% (Figure 4B), 98% (Figure 4C), 99% (Figure 4D), 100% (Figure 4E), 93% (Figure 4F), 98% (Figure 4H), 98% (Figure 5A), 98% (Figure 5B), 95% (Figure 5C), 94% (Figure 5D), and 93% (Figure 5E). The only meta- analysis with an I2 value below the 65% cutoff value was miR-26-5p, with heterogeneity at 38% (Figure 4G). One reason for the considerable heterogeneity could be the different normalization strategies used in the different studies; the normalizers in the original studies were U6, RNU48, Cel-miR-39, UniSP6, miR-484, miR-16, miR-103-3p, and global mean based on (qPCR) array data. Some studies did not disclose which normalization strategy they used (Appendix A).

## 4. Discussion

Changes in miRNA expression seem to be an important factor in cancer development, and miRNAs were demonstrated to play a role in the occurrence, migration, and invasion of tumors [77,78,79,80,81,82]. Cancer cells also release RNAs into the circulation, and it is, therefore, possible that specific miRNAs can be novel biomarkers for the early diagnosis of OC. Hence, many studies on the diagnostic significance of circulating miRNA emerged [83]. Thus, with the aim to perform a synthesis of the currently available knowledge of circulating RNAs in relation to OC diagnosis, 48 studies were included in the current analysis. After data extraction, 23 studies were eligible for meta-analysis. Thirteen different miRNAs were entered into for the meta-analyses; nine miRNAs were identified as upregulated in OC patients compared to controls (miR-145-5p, -205-5p, -21-5p, -328-5p, -125-5p, -141-5p, -200b-3p and -200c-3p). The between-study heterogeneity was considerable, with miR-26 as the only exception. Methodological differences might cause substantial heterogeneity between studies. For example, it could be due to different cancer subtypes between studies. Unfortunately, there were not sufficient original studies to perform subtype analysis to test this hypothesis (Appendix A). Another reason for the observed heterogeneity could be different normalization strategies used in individual studies [19] There is no generally accepted approach for analyzing the results from measurements of circulating RNAs related to biomarker identification, making it a challenge to compare studies. Different study designs might also increase heterogeneity between studies; the approaches to identifying the different miRNAs vary; some use a hypothesis-free approach using either sequencing or qRT-PCR array before validating their results in a larger population. The Kim et al. study (investigating miR-21-5p, -93-5p, -145-5p and miR-200c-3p) is the only study that used women with benign tumors or cysts as controls and not healthy women, although the change in type of control did not lead to Kim et al. being an outlier [33].

The miR-200 family members were the most frequently studied miRNAs; however, the meta-analyses indicated a high level of heterogeneity between studies and study variability, suggesting that the miR-200 family might constitute the best candidates for a novel biomarker for early OC diagnosis. However, it might be beneficial to combine multiple OC-associated miRNAs from the same blood sample; this might provide a more robust overall biomarker. Sensitivity might be improved using ratios of circulating miRNAs rather than single miRNAs. For example, one could envision testing the ratio between miR-205-5p, increased in OC, and miR-26-5p, reduced in OC patients. The ratio between these is expected to be significantly increased with a large effect size in OC patients compared to controls. 

The subgroup analyses were performed on both plasma and serum for miR-200c-3p studies, and based on these, miR-200c-3p was identified as increased in ovarian cancer compared to controls. The plasma studies had a slightly lower increase than the serum studies. This suggests there are no sizeable biological differences between serum and plasma studies. Numerous studies were conducted on miRNAs in OC and other solid tumor disease sites. MiRNAs are particularly appealing as diagnostic biomarkers for early-stage disease because of their stability in circulation [75]. Some studies used animal models such as xenografts to evaluate the potential of the miRNAs and assess the origin of the miRNAs in the circulation; i.e., miR-200c and -200a were identified in serum in a mouse xenograft model [75]. 

The members in the miR-200 family were identified in a wide range of cancers, and investigated for their effect on the different hallmarks of cancer [84]. Furthermore, epithelial-mesenchymal transition (EMT) and zinc-finger E-box binding homeobox (ZEB) are two known targets of the miR-200 family [85]. MiR-200a (osteosarcoma [86]), miR-200b (gastric cancer, oral squamous cell carcinoma, osteosarcoma [87,88]), miR-200c (prostate cancer [89,90]), and miR-429 (cervical cancer, osteosarcoma, thyroid cancer [91,92,93]) all target ZEB1/2 in different cancers. MiR-205 was identified to regulate the EMT pathway by targeting TP53 (P53) [94] and ZEB1 [95]. Moreover, miR-205 was associated with granulosa cell apoptosis and estradiol synthesis through targeting cAMP response element-binding protein 1 (CREB1) [96]. MiRNA-145 was identified in association with a wide range of cancers, and found to have a different regulatory functions [97]. It functions as a tumor suppressor, by targeting multiple genes including c-Myc and VEGF [98]. MiR-21 is extensively studied in various biological processes and diseases [99,100,101,102]. It was identified to function as a down regulator of PTEN, SMAD7 and PDCD4 [103,104,105,106]. PTEN is also a target for miR-328 [107]; however, miR-328 expression is associated with several cancers and targets [108]. P53 was identified in correlation to several miRNAs, e.g., miR-125b, which was found to suppress p53 function [109].

While both miRNAs and CA125 showed potential as diagnostic biomarkers for ovarian cancer, miRNAs were found to have higher sensitivity and specificity compared to CA125 [9]. Additionally, miRNAs have the advantage of being stable in blood samples and are less likely to be influenced by factors such as age or menstrual cycle, making them a more reliable biomarker [110,111]. Combining multiple biomarkers, such as miRNAs, CA125, and HE4, can increase the sensitivity and specificity of ovarian cancer diagnosis. By using a panel of biomarkers instead of a single marker, the accuracy of diagnosis can be improved, potentially leading to earlier detection and better patient outcomes.

While the field of miRNA biomarker research is immature and still developing, our meta-analysis clearly showed that levels of specific circulating miRNAs were significantly associated with OC diagnosis. These findings were obtained, despite the challenge that meta-analysis included a relatively low number of studies investigating the same miRNAs; most of the investigated miRNAs were only investigated in one study (Figure 2). Considering this, the current analysis showed that some of the identified miRNAs might have the potential for development as novel diagnostic biomarkers for OC, whether alone or in combination. MiR-145, and -205 had more than 2-fold higher levels in the circulation of OC patients compared with control participants. Hence, these showing miRNAs demonstrate good potential to be novel biomarkers for detecting OC. However, the inter-study heterogeneity was considerable. Another limitation of this review and meta-analysis was that not all identified studies could be included due to the lack of displayed extractable data, which limits the strength of the meta-analysis. Furthermore, additional and/or other miRNAs might be associated with OC than those investigated in the current meta-analyses, because this review focused on miRNAs identified in three or more studies. 

## 5. Conclusions

In conclusion, specific miRNAs might have potential as diagnostic biomarkers in OC; in particular, miR-145, and miR-205 as well as miRNAs from the miR-200 family: miR-141-5p, and -200c-3p. However, published studies of circulating miRNAs generally investigated few subjects, lack consensus on methodological approaches, and findings were often not confirmed by other studies. Although these considerations limit the possibility for firm conclusions, our analysis did identify specific circulating miRNAs with consistent associations with OC, which should be investigated further for their potential as biomarkers for early diagnosis of OC.

## Figures and Tables

**Figure 1 biomolecules-13-00871-f001:**
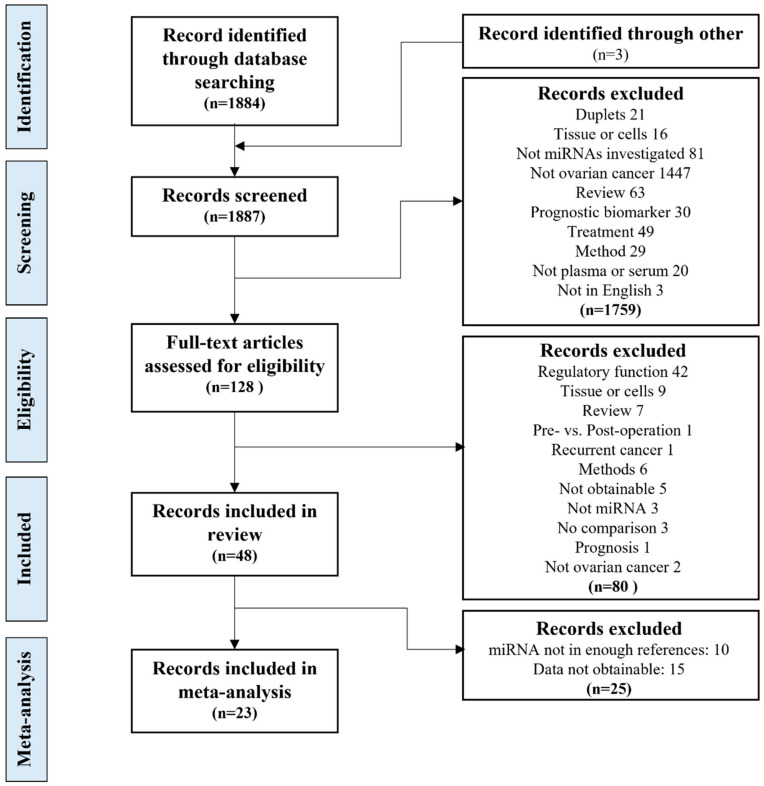
The study selection process. The study selection process, from the database search, screening and eligibility evaluation, inclusion for the systematic review and meta-analysis, including the exclusion criteria.

**Figure 2 biomolecules-13-00871-f002:**
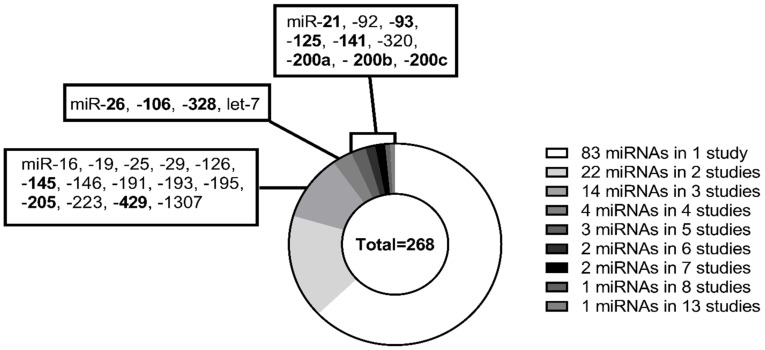
Distribution of miRNAs in included studies. Two hundred sixty-eight miRNAs were identified through the screening of 48 articles. In one study, 83 miRNAs were identified, 22 miRNAs were identified in two studies, 14 miRNAs in three studies, 4 miRNAs in four studies, 3 miRNAs in five studies, and 2 miRNAs in six and seven studies. Additionally, one miRNA in eight and 13 studies. The miRNAs highlighted in bold are the miRNAs that were included in the meta-analysis.

**Figure 3 biomolecules-13-00871-f003:**
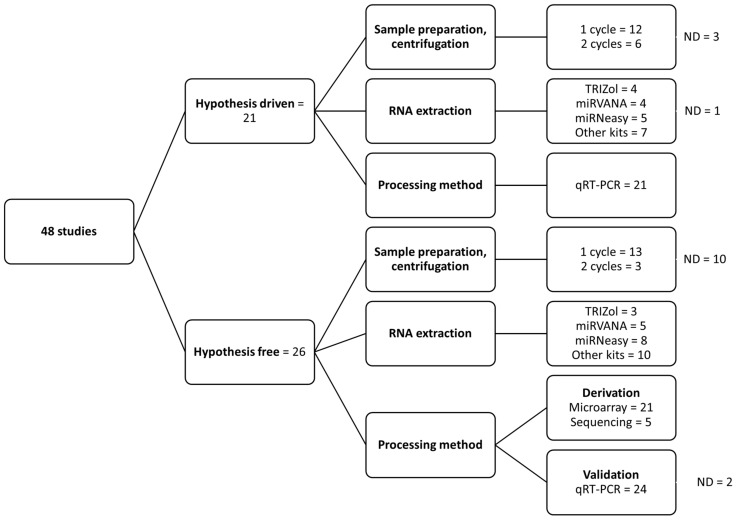
Summary of pre-analytical factors in the included studies. Summary of difference in the methodology used for processing samples and RNA extraction. ND, not defined in the study.

**Figure 4 biomolecules-13-00871-f004:**
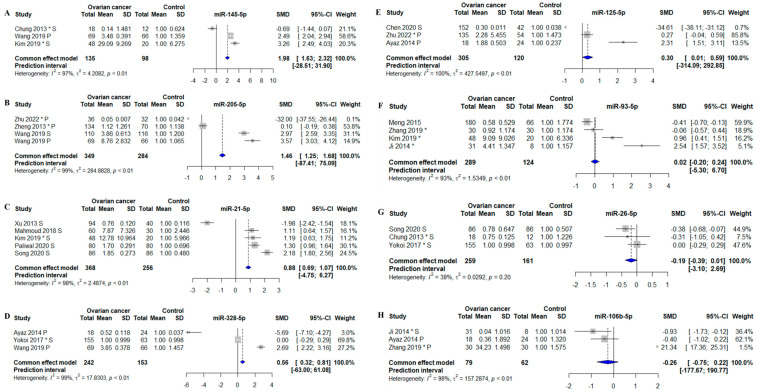
Meta-analysis of studies investigating miR-145-5p, -205-5p, -21-5p, -328-5p, -125-5p, -93-5p, -26-5p, and -106-5p. (**A**) Forest plot for miR-145-5p, the efficiency for miR-145-5p to distinguish between OC pts and controls. (**B**) Forest plot for miR-205-5p. (**C**) Forest plot for miR-21-5p. (**D**) Forest plot for miR-328-5p. (**E**) Forest plot for miR-125-5p. (**F**) Forest plot for miR-93-5p. (**G**) Forest plot for miR-26-5p. (**H**) Forest plot for miR-106-5p. CI, confidence interval; SD, standard deviation; SMD, standardized mean difference. *, Data extracted using web plot digitizer.

**Figure 5 biomolecules-13-00871-f005:**
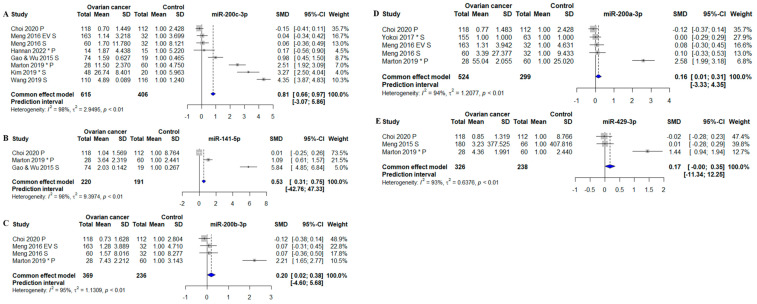
Meta-analysis of studies investigating microRNAs belonging to the miR-200 family (miR-141-5p, miR-200a-3p, miR-200b-3p, miR-200c-3p, and miR-429-3p). (**A**) Forest plot for miR-200c-3p, the efficiency for miR-200-3p c to distinguish between OC pts and controls. (**B**) Forest plot for miR-141-5p. (**C**) Forest plot for miR-200b-3p. (**D**) Forest plot for miR-200a-3p. (**E**) Forest plot miR-429-3p. CI, confidence interval; SD, standard deviation; SMD, standardized mean difference. *, Data extracted using web plot digitizer.

**Figure 6 biomolecules-13-00871-f006:**
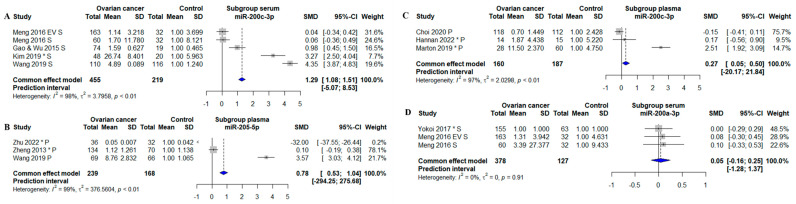
Subgroup meta-analysis of studies investigating. (**A**) Forest plot of a subgroup for serum based miR-200c-3p studies, the efficiency for miR-200c-3p to distinguish between OC pts and controls. (**B**) Forest plot of a subgroup for plasma based miR-205-5p studies. (**C**) Forest plot of a subgroup for plasma based miR-200c-3p studies. (**D**) Forest plot of a subgroup for plasma based miR-200a-3p studies. CI, confidence interval; SD, standard deviation; SMD, standardized mean difference. S, serum. P, plasma. *, Data extracted using web plot digitizer.

**Table 1 biomolecules-13-00871-t001:** Characteristics of studies included in the meta-analysis. The full table can be found in Appendix A. An asterisk (*) means that data were extracted using a web plot digitizer. EOC = epithelial ovarian cancer- OC = ovarian cancer. HGSOC = high-grade serous ovarian cancer. BOT = borderline ovarian tumor. SEOC = Serous epithelial ovarian cancer. ND = not disclosed.

Ref	Nationality	Study Population Cases/ Controls	Sample Size Cases/ Controls	Age Cases/Controls	Plasma/Serum	miRNA Investigated
[27]	Turkey	OC/healthy	18/24	58.88 ± 16.77/53.73 ± 9.42	Plasma	-19-3p, -30a-5p, 34c-5p, -106b-5p, 150-5p, 191-5p, -206-3p, -320a-3p, -548a-3p, 574-3p, 590-5p, -645-3p
[38]	China	OC/healthy	152/42	50.8/ND	Serum	-125b-5p
[49]	United States, China	OC/healthy	118/112	(Medians) 57/55	Serum, plasma	-141-5p, -429-3p, -200a/b/c-3p
[60] *	Korea	EOC/healthy	18/12	(Median) 57.5	Serum	-26a-5p, -132-3p, -143-3p, -145-5p, let-7b-5p
[73]	China	EOC/healthy	93/50	ND	Serum	-141-5p, -200c-3p
[29] *	Australia	HGSOC/benign	14/15	62.9/63	Plasma	-200c-3p
[31] *	China	OC/healthy or benign	31/8 healthy, 23 benign	ND	Serum	-22-3p, -93-5p, -106b-5p, -451-5p
[33] *	Korea	HGSOC and Non-HGSOC/benign or BOT	HGSOC 39 Non-HGSOC 9/10 benign, 10 BOT	HGSOC 58.1 Non-SOC 43.7/Benign 57.2 BOT 41.6	Serum	-21-5p, -93-5p, 200c-3p
[36]	Egypt	OC/healthy	60/30	55.5/ND	Serum	-21-5p
[37] *	Hungary	OC/healthy	28/60	57.03 ± 9.56/ 56.17 ± 12.18	Plasma	-34a/b/c, -141-5p, -429-3p, -200a/b/c-3p
[39]	Germany	EOC/healthy	180/66	60/59	Serum	-25, -93-5p, -429-3p
[40]	Germany	EOC/healthy, benign	60/32 Healthy, 20 benign	(medians) 56/Healthy 56, Benign 46	Serum	-200a/b/c-3p
[41]	Germany	EOC/healthy, benign	163/32 Healthy, 20 benign	(medians) 60/Healthy 56, Benign 46	Serum	-200a/b/c-3p
[43]	India	EOC/healthy	80/80	ND	Serum	-21-5p, -22-3p
[51]	China	OC/healthy	86/86	46.1 ± 2.8/46.8 ± 2.6	Serum	-21-5p, -26b-5p
[55]	China	OC/healthy	101/100	ND	Plasma	-145-5p, -205a-5p, -328-3p, 346-5p
[56]	China	OC/healthy	110/116	ND	Serum	-127-3p, -143-3p, -205-5p, 346-5p, -200c-3p
[58]	China	EOC/healthy	94/40	59.2 ± 6.8	Serum	-21-5p
[59] *	Japan	OC/healthy	155/63	54.1/54.1	Serum	-26a-5p, -130b-3p, -142-3p, -328-3p, -374a-5p, -766-3p, -200a-3p, Let-7d-5p
[62] *	China	OC/healthy	30/30	53.34 ± 2.34/52.67 ± 3.21	Plasma	-93-5p, -99b-5p, -106a-5p, -122-5p, -185-5p, -let-7d-5p
[74] *	China	EOC/healthy	360/200	53.22 ± 10.38	Plasma	-205-5p, let-7f-5p
[64] *	China	EOC/healthy	135/54	54/55	Serum	-125b-5p
[65] *	China	OC/healthy	36/32	ND/ND	Plasma	-205-5p

## Data Availability

All data are available in the main article or in the Appendix A.

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
