# Peer review of "Circulating microRNAs for Early Diagnosis of Ovarian Cancer: A Systematic Review and Meta-Analysis"

_biomolecules, 2023, doi:10.3390/biom13050871_

Round 1
Reviewer 1 Report
In this study, the authors have conducted a systematic review and meta-analysis to summarize and evaluate the potential of different circulating miRNAs as an early diagnostic biomarker for OC. Although overall the manuscript is well written it may be improved and can be made more infomative. The authors may also provide infomation if avaliable as to whether any other established biomarkers for ovarian cancers were also studied in some of these studies. This way the advantage of circulating miRNAs over these biomarkers or in combination can be assessed. Also miRNAs exist in different forms ( a , b, -3p -5p etc), so referring to miRNA alone without specific forms may not be as infomative. It is important that authors provide such infomation in discussion and conclusion to be meaningful as guide to future studies. They should also discuss possible targets of the miRNAs and potential targets. Addtionally they should also discuss circulating miRNA profiles if reported in animal models of ovarian cancers. Such infomation may be useful for focusing on specific miRNAs found in human studies for their relevance as biomarkers.
Author Response
Answers to questions from reviewers
Reviewer #1
In this study, the authors have conducted a systematic review and meta-analysis to summarize and evaluate the potential of different circulating miRNAs as an early diagnostic biomarker for OC. Although overall the manuscript is well written it may be improved and can be made more infomative.
Answer: Thank you very much for the constructive comments; we have strived to improve according to your suggestions.
The authors may also provide infomation if avaliable as to whether any other established biomarkers for ovarian cancers were also studied in some of these studies. This way the advantage of circulating miRNAs over these biomarkers or in combination can be assessed.
Answer: Thank you for the suggestion. We have investigated in the included studies if the cancer biomarkers CA125 and HE4 were compared to the levels of miRNAs. However, only 28 out of the included 48 studies also described CA125 and/or HE4 biomarker levels. We have compared reported CA125 levels with identified miRNAs, and studies report generally better discriminatory capacities when combining CA125 with specific miRNA biomarkers. (Lines 191 to 199) We have incorporated this in the discussion (lines 397 to 405) and added it to the supplementary data file S1 columns U and V.
Also miRNAs exist in different forms ( a , b, -3p -5p etc), so referring to miRNA alone without specific forms may not be as informative. It is important that authors provide such information in discussion and conclusion to be meaningful as guide to future studies.
Answer: We agree with this consideration, and during our data extraction process, we obtained this information. Information about the isoforms (a, b, c) is already given in the review. However, we have clarified our data extraction procedure regarding miRNA arms (throughout the manuscript). Thus, we have added information about the -3p or -5p arms of the miRNAs covered in the systematic review (figures 4, 5, and 6). Information about miRNA isoform -3p and -5p arms were extracted and can be found in the supplementary file 1. For some of the articles, information of -3p and -5p arms of the investigated miRNAs was not given; in these cases, the doubts were resolved using the miRNA database miRbase.org as now detailed in the methods section (lines 105 to 110).
They should also discuss possible targets of the miRNAs and potential targets.
Answer: For the significant miRNAs we have included this in the discussion (lines 380 to 396).
Additionally, they should also discuss circulating miRNA profiles if reported in animal models of ovarian cancers. Such infomation may be useful for focusing on specific miRNAs found in human studies for their relevance as biomarkers.
Answer: We agree that information about the origin of the altered miRNA in circulation is important. To gain insight into this, we have searched for and included information regarding circulating miRNAs of ovarian cancer xenografts in mouse models; we included this in the discussion (lines 375 to 379).

Reviewer 2 Report
In this manuscript the authors performed a meta-analysis of specific circulating miRNAs in ovarian cancer, post-transcriptional regulator control mRNA translational inhibition or degradation, suggesting miR-145, miR-205, miR-141, and miR-200 as potential diagnostic biomarkers of the disease. The authors focused on a specific time period (15 months). I have some comments here:
Introduction is limited and could be strengthen more
The authors should explain the use of Newcastle-Ottawa Scale score in their assessment, discussing the advantages compared with similar approaches.
The purpose of pre-analytical factors in this method could also be clarified in more detail. Any biological difference between serum and plasma evidences?
Did the authors perform a robustness analysis to evaluate the result reliability?
Also, did they investigate the publication bias influence using for example Deeks' funnel plot asymmetry test
It’s a bit difficult to understand forest plots, the x-axis is sensitivity, specificity?
Author Response
Answers to questions from reviewers
Reviewer #2
In this manuscript the authors performed a meta-analysis of specific circulating miRNAs in ovarian cancer, post-transcriptional regulator control mRNA translational inhibition or degradation, suggesting miR-145, miR-205, miR-141, and miR-200 as potential diagnostic biomarkers of the disease. The authors focused on a specific time period (15 months). I have some comments here:
Answer: Thank you for a thorough review. We would like to address the comment about the specific time period of the review. The review has included all published original articles up until February 2023 and is therefore not limited to a specific time period.
Introduction is limited and could be strengthen more
Answer: We did initially try to keep the introduction concise. However, we value the opportunity to expand the introduction, where we have added information about the limitation within the biomarker development field. (Lines 69 to 75). We also commented on this in the discussion (lines 406 to 408).
The authors should explain the use of Newcastle-Ottawa Scale score in their assessment, discussing the advantages compared with similar approaches.
Answer: The Newcastle-Ottawa Scale is a newer scale for assessing the quality of non-randomized studies in meta-analyses, (https://www.ohri.ca/programs/clinical_epidemiology/oxford.asp). The advantage of the NOS scale is that it specifically relates to non-randomized studies, which most biomarker studies belong to, whereas another popular appraisal tool, the Cochrane risk of bias tool vs 2, is specifically targeted individually randomized studies, which would not be appropriate here. We have added this information in the methods section (lines 113 to 116).
The purpose of pre-analytical factors in this method could also be clarified in more detail. Any biological difference between serum and plasma evidences?
Answer: Thank you for a very relevant suggestion, we performed a sub-group analysis on miR-200a-3p, -200c-3p, and -205-5p because these categories contained sufficient included articles to perform sub-group analysis. Moreover, we have added S and P to all studies in all meta-analyses to clarify which blood component has been used for the study. (Figures 4, 5, and 6. Lines 286 to 297 and 370 to 375)
Did the authors perform a robustness analysis to evaluate the result reliability? Also, did they investigate the publication bias influence using for example Deeks' funnel plot asymmetry test
Answer: Thank you for your very relevant questions. Yes, we performed a Cook’s distance analysis to identify any outliers, and we did not identify any outliers. Funnel plots were also created, and some asymmetry to the plots was observed (Lines 298 to 307 supplementary figure S1).
It’s a bit difficult to understand forest plots, the x-axis is sensitivity, specificity?
Answer: The forest plot is a standardized graphical representation; The X-axis depicts the Standardized Mean Difference (SMD) for OC and C (controls) for each study. With all control groups calculated and set to 1 (100%), the fold-change for the corresponding OC-group would always be 1 more than the calculated SMD (i.e. SMD = Fold-change-1 for each study included in the Forest plot).

Round 2
Reviewer 1 Report
The authors have revised the manuscript to the satistaction of the reviewer and thus the reviewer recommends its publication in Biomolecules.